# STIL binding to Polo-box 3 of PLK4 regulates centriole duplication

**Christian Arquint[†], Anna-Maria Gabryjonczyk[†], Stefan Imseng[†], Raphael Böhm[†], Evelyn Sauer, Sebastian Hiller, Erich A Nigg, Timm Maier***

Biozentrum, University of Basel, Basel, Switzerland

**Abstract** Polo-like kinases (PLK) are eukaryotic regulators of cell cycle progression, mitosis and cytokinesis; PLK4 is a master regulator of centriole duplication. Here, we demonstrate that the SCL/TAL1 interrupting locus (STIL) protein interacts via its coiled-coil region (STIL-CC) with PLK4 in vivo. STIL-CC is the first identified interaction partner of Polo-box 3 (PB3) of PLK4 and also uses a secondary interaction site in the PLK4 L1 region. Structure determination of free PLK4-PB3 and its STIL-CC complex via NMR and crystallography reveals a novel mode of Polo-box–peptide interaction mimicking coiled-coil formation. In vivo analysis of structure-guided STIL mutants reveals distinct binding modes to PLK4-PB3 and L1, as well as interplay of STIL oligomerization with PLK4 binding. We suggest that the STIL-CC/PLK4 interaction mediates PLK4 activation as well as stabilization of centriolar PLK4 and plays a key role in centriole duplication.

## Introduction

Centrosomes are the major organizing centers for the microtubule network in animal cells and facilitate many microtubule-dependent cellular processes throughout the cell cycle (reviewed in *Bettencourt-Dias and Glover, 2007*; *Bornens, 2012*; *Gonczy, 2012*). During interphase, centrosomes contribute to cell shape, motility and polarity; in mitosis, they form the poles of the mitotic spindle and direct chromosome segregation. The core components of centrosomes, the centrioles, also function as basal bodies for the assembly of cilia and flagella. Mutations of centrosomal proteins have been associated with a variety of human diseases, notably ciliopathies, microcephalies and dwarfisms (*Nigg and Raff, 2009*; *Chavali et al., 2014*). Furthermore, numerical and/or structural centrosome abnormalities have been implicated in carcinogenesis (*Nigg, 2002*; *Basto et al., 2008*; *Ganem et al., 2009*; *Nigg and Raff, 2009*).

Centrosomes duplicate during S-phase by formation of a procentriole, the daughter centriole, orthogonally arranged to each pre-existing mother centriole (*Firat-Karalar and Stearns, 2014*; *Sluder, 2014*). The duplication process relies on a set of proteins conserved from *Caenorhabditis elegans* and *Drosophila melanogaster* to humans. The human core components are: the serine/threonine Polo-like kinase PLK4 (ZYG-1 in *C. elegans*), the centrosomal protein Cep192 (SPD-2 in *C. elegans*), the spindle assembly 6 homolog (*C. elegans*) protein SAS-6, the SCL/TAL1 interrupting locus protein STIL (SAS-5 in *C. elegans*) and the centrosome protein CPAP (SAS-4 in *C. elegans*) (*Strnad and Gonczy, 2008*; *Azimzadeh and Marshall, 2010*; *Carvalho-Santos et al., 2011*; *Nigg and Stearns, 2011*; *Brito et al., 2012*; *Gonczy, 2012*). Overexpression of either, PLK4, SAS-6 or STIL, causes formation of multiple daughter centrioles around a single mother centriole (*Habedanck et al., 2005*; *Kleylein-Sohn et al., 2007*; *Strnad et al., 2007*; *Tang et al., 2011*; *Arquint et al., 2012*; *Vulprecht et al., 2012*), while depletion of any one of these proteins blocks centriole formation (*Bettencourt-Dias et al., 2005*; *Habedanck et al., 2005*; *Leidel et al., 2005*; *Tang et al., 2011*; *Arquint et al., 2012*; *Vulprecht et al., 2012*). Thus, PLK4, SAS-6 and STIL constitute key centriole duplication factors, the activity and levels of which need to be tightly controlled to maintain the

*For correspondence: timm.maier@unibas.ch

[†]These authors contributed equally to this work

**Competing interests:** The authors declare that no competing interests exist.

**eLife digest** Centrioles are structures that organize the molecular scaffolding inside cells, which is important for a cell's shape and activity, as well as the segregation of duplicated chromosomes during cell division. Centrioles also form part of the base of the antenna-like structures called cilia, which project out from the cell's surface and allow cells to sense chemicals and touch or even to move. A cell that is not dividing contains a pair of centrioles. In dividing cells, the two centrioles duplicate once per cycle of division and a new centriole forms next to each of the existing ones. It is essential that centrioles duplicate only once, because extra copies can lead to problems that may cause birth defects and cancer.

Centrioles require two proteins, called PLK4 and STIL, in order to duplicate. An excess of either of these proteins results in extra centrioles. On the other hand, if these are missing, duplication cannot take place. PLK4 belongs to a large family of enzymes called kinases. A kinase attaches a phosphate group to other proteins, which can either activate or deactivate the other protein. PLK4 can add phosphate groups onto STIL, but it is not known precisely how these two proteins interact with each other.

Arquint, Gabryjonczyk, Imseng, Böhm et al. have analyzed this interaction in human cells and found that PLK4 and STIL bind directly to one another. Part of the STIL protein adopts a so-called 'coiled-coil' structure in which twisted lengths of protein wrap around each other like a piece of string. The coiled-coil interacts with two different parts of PLK4.

Following on from these observations, the three-dimensional structure of PLK-4 bound to STIL was visualized using X-ray crystallography and nuclear magnetic resonance. These techniques revealed that the coiled-coil region of STIL forms an elongated structure and PLK-4 interacts along its entire length.

Arquint, Gabryjonczyk, Imseng, Böhm et al. then analyzed whether PLK4 and STIL need one another in order to get recruited to centrioles. When PLK4 was depleted in cells, STIL was lost from centrioles, suggesting that PLK4 directly recruits STIL. However, contrary to expectations, when STIL levels were reduced, PLK4 accumulated at centrioles. This suggests that STIL maintains appropriate levels of PLK4 via stimulation of its kinase activity. Further work is needed to precisely understand how PLK4 and STIL interact with other proteins that act downstream to lead to the formation of new centrioles in a highly controlled manner.

correct centriole number (*Strnad et al., 2007*; *Guderian et al., 2010*; *Holland et al., 2010*; *Arquint and Nigg, 2014*). In addition to these core components, other proteins are also essential for centriole duplication in human cells, this includes notably Cep152 (*Cizmecioglu et al., 2010*; *Dzhindzhev et al., 2010*; *Hatch et al., 2010*), which cooperates with Cep192 in PLK4 recruitment (*Kim et al., 2013*; *Sonnen et al., 2013*; *Park et al., 2014*).

The early phase of centriole biogenesis is marked by the assembly of the cartwheel structure that serves as a scaffold for deposition of centriolar microtubules and confers the characteristic ninefold symmetry to the centriole (*Nakazawa et al., 2007*; *Gonczy, 2012*; *Hirono, 2014*; *Winey and O'Toole, 2014*). SAS-6 has been shown to self-assemble into cartwheel-like structures in vitro, indicating that it is a central component of the cartwheel (*Kitagawa et al., 2011*; *van Breugel et al., 2011*; *Guichard et al., 2013*; *van Breugel et al., 2014*). In human cells, SAS-6, STIL and PLK4 localize to the cartwheel region, suggesting a functional interaction of these proteins in cartwheel assembly (*Strnad et al., 2007*; *Arquint et al., 2012*; *Sonnen et al., 2012*; *Fong et al., 2014*). Such an interaction is supported by recent evidence demonstrating that PLK4 regulates complex formation between STIL and SAS-6 via phosphorylation of STIL (*Dzhindzhev et al., 2014*; *Ohta et al., 2014*; *Kratz et al., 2015*). This process depends on two highly conserved regions of STIL: a short coiled-coil (CC) motif (STIL-CC, residues 720–751) and the STAN (STIL/Ana2) domain (residues 1061–1147) (*Stevens et al., 2010*; *Dzhindzhev et al., 2014*). This recent progress focuses attention on a detailed mechanistic understanding of the interaction between STIL and PLK4, and this in turn requires definitive structural information.

PLK4 belongs to the PLK family, which in vertebrates comprises four functional paralogues, PLK1-4. PLKs are characterized by an N-terminal Ser/Thr- kinase domain followed by a C-terminal region

containing two or three Polo-box folds (PB), which regulate substrate binding, kinase activity, and localization (reviewed in *Lowery et al., 2005*; *Archambault and Glover, 2009*; *Zitouni et al., 2014*). Among the PLKs, PLK1 is the best studied; it comprises two Polo-boxes, PB1 and PB2, that form a Polo-box domain (PBD), through intramolecular heterodimerization. The PLK1-PBD generally binds to target proteins after their phosphorylation on Ser/Thr- sites within a PBD-docking motif (*Cheng et al., 2003*; *Elia et al., 2003a*, *2003b*; *Yun et al., 2009*; *Xu et al., 2013*); however, in the context of the *Drosophila* microtubule-associated protein Map205 phospho-independent binding has also been described (*Archambault et al., 2008*). PLK4 is unique among the PLKs as it contains three -rather than two- Polo-boxes (PB1-3) (*Slevin et al., 2012*). The first two Polo-boxes of PLK4, PB1 and PB2 (formerly referred to as cryptic Polo-box [CPB]), are sufficient for centriole localization of PLK4 (*Habedanck et al., 2005*; *Slevin et al., 2012*). Isolated PLK4-PB3 can also localize to centrioles, but with less efficiency (*Leung et al., 2002*; *Slevin et al., 2012*). In contrast to PLK1-PBD, PLK4-PB1/2 as well as PB3 have been described to form intermolecular homodimers and to bind their targets in a different, phospho-independent manner (*Leung et al., 2002*; *Slevin et al., 2012*; *Kim et al., 2013*; *Park et al., 2014*; *Shimanovskaya et al., 2014*). Recent work has established a crucial role for the binding of acidic regions in Cep192 and Cep152 to basic residues in PLK4-PB1/2 (*Kim et al., 2013*; *Sonnen et al., 2013*; *Park et al., 2014*). However, no interactions of PLK4-PB3 with binding partners have been resolved so far. Moreover, the relevance of the reported domain-swapped structure of murine PB3 (*Leung et al., 2002*) for in vivo interactions remains unclear.

Here, we identify STIL as a direct interaction partner and substrate of PLK4 and confirm that the STIL-CC region is essential for STIL function in centriole duplication. Most importantly, we determined the solution structure of the human PLK4-PB3 and a crystal structure of the PLK4-PB3/STIL-CC complex and use structure-based mutagenesis of STIL to demonstrate an essential role of STIL-CC for PLK4 binding and the regulation of centriole biogenesis in vivo. Specifically, we show that STIL-CC interacts with two regions within PLK4: it targets not only the L1 region but also is the first identified binding partner of the unique PLK4-PB3. We further show that STIL-CC binding is implicated in the stabilization of centriolar PLK4 and its concomitant activation. Collectively, our results contribute to a detailed structural and mechanistic understanding of a crucial initial step of centriole biogenesis.

## Results

### PLK4 and STIL interact in vivo to regulate centriole duplication

To identify centrosomal binding partners of the PLK4 Polo-box motifs, we performed an S-peptide pulldown experiment coupled to mass spectrometry analysis. We generated a U2OS Flp-In T-REx cell line that allowed for inducible expression of an S-peptide-EGFP-tagged PLK4 fragment (residues 570–970) comprising the three Polo-boxes PB1-3. We identified a set of centrosomal proteins including the two well-known PLK4-PB1/2 binding partners Cep152 and Cep192 (16 and four identified peptides, respectively) (*Cizmecioglu et al., 2010*; *Dzhindzhev et al., 2010*; *Hatch et al., 2010*; *Kim et al., 2013*; *Sonnen et al., 2013*). In addition, the key centriole duplication factor STIL co-purified with the PLK4 fragment (one identified peptide) (*Firat-Karalar et al., 2014*). This prompted us to further analyze the functional and structural interaction between PLK4 and STIL.

As 3D-SIM imaging of U2OS cells revealed extensive co-localization of STIL and PLK4 at the proximal end of daughter centrioles (*Figure 1A*), we asked whether the two proteins depend on each other for recruitment to this site. Upon depletion of PLK4, localization of STIL to centrioles was drastically reduced (1.7 ± 2.3% residual intensity compared to untreated cells, *Figure 1B*), suggesting that PLK4 is essential for STIL centriolar targeting and/or maintenance. On the other hand, PLK4 localization to centrioles was not abrogated in STIL depleted cells. On the contrary, centriolar PLK4 levels were strongly elevated and PLK4 localized in a ring-, rather than a spot-like pattern to the outer wall of centrioles (*Ohta et al., 2014*) (*Figure 1C*). Western blot analysis confirmed significant elevation of PLK4 levels in STIL depleted cells (*Figure 1—figure supplement 1A–D*). This increase in PLK4 levels was comparable to that observed after depletion of βTrCP, which is known to interfere with PLK4 degradation (*Cunha-Ferreira et al., 2009*; *Rogers et al., 2009*; *Guderian et al., 2010*; *Holland et al., 2010*) (*Figure 1—figure supplement 1*). These data suggest that PLK4 degradation is strongly reduced in the absence of STIL, which then results in its accumulation around centrioles. In further support of a functional interaction between PLK4 and STIL, we also observed that STIL was

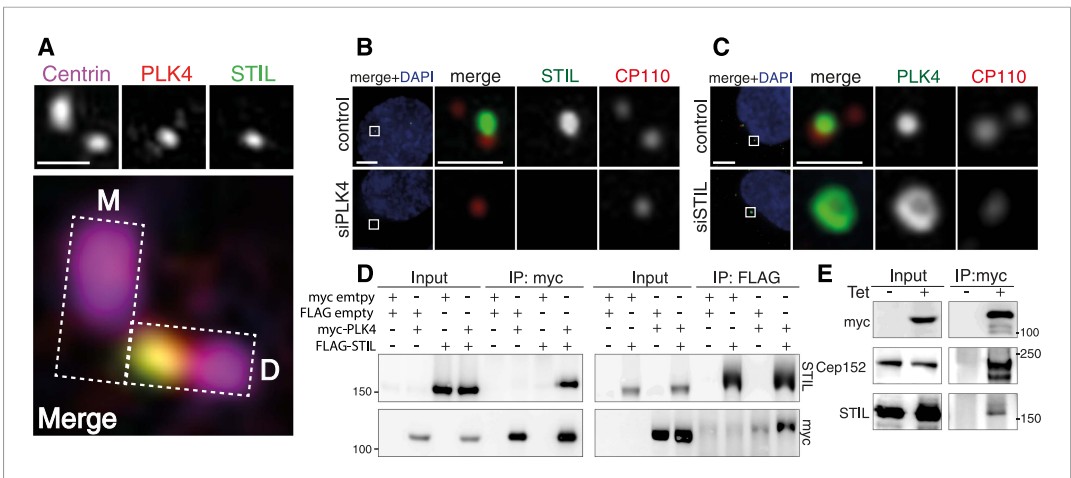

**Figure 1**. STIL is an interaction partner of PLK4. (**A**) U2OS cells were fixed and stained with the indicated antibodies for 3D-SIM imaging. A representative 3D-SIM image is shown, demonstrating the co-localization of PLK4 and STIL at the daughter centriole. Top panel: centrin (purple), PLK4 (red), STIL (green). Scale bar: 0.5 μm. Bottom panel: magnified view of the centrosome (overlay image). The rectangles illustrate the orientation of the mother (M) and daughter (D) centrioles. (**B**) Immunostaining of STIL localization in U2OS cells depleted of endogenous PLK4. Cells were transfected for 72 hr with control or PLK4 siRNA oligonucleotides and stained with the indicated antibodies. DAPI is shown in blue. Scale bars: 1 μm. (**C**) Immunostaining of PLK4 localization in U2OS cells depleted of endogenous STIL (control and siSTIL). 'Material and methods' as in (**B**). In (**B**) and (**C**), only prophase cells harboring 1–2 centrioles were analyzed (indicative of successful PLK4 or STIL depletion, respectively). (**D**) Western blots showing the interaction of myc-tagged PLK4 and FLAG-tagged STIL in HEK293T cells. Cells were transfected with the indicated plasmids for 36 hr, followed by lysis and immunoprecipitation using anti-myc or anti-FLAG antibodies. Antibodies used for Western blot detection are indicated. (**E**) Western blot showing the interaction of myc-PLK4 with endogenous STIL. Myc-PLK4 expression was induced by addition of tetracycline to U2OS T-REx cells stably harboring the myc-PLK4 transgene (± Tet, 24 hr). Cells were processed for anti-myc co-immunoprecipitations and Western blot analysis using the indicated antibodies.

The following figure supplements are available for figure 1:

**Figure supplement 1**. Plk4 levels are elevated in STIL depleted cells.

**Figure supplement 2**. STIL is a phosphorylation target of PLK4 and PLK4-ND.

---

phosphorylated by a recombinant GST-PLK4$_{1-430}$ fusion protein in vitro (*Dzhindzhev et al., 2014*; *Ohta et al., 2014*; *Kratz et al., 2015*) (*Figure 1—figure supplement 2A*).

To confirm the interaction of STIL with PLK4, we transfected HEK293T cells with myc- and FLAG-tagged versions of both proteins and performed co-immunoprecipitation experiments. As expected, we found STIL and PLK4 to be present in the immunoprecipitates of the respective interaction partner (*Figure 1D*). Moreover, we detected endogenous STIL along with Cep152 in an immunoprecipitate of myc-PLK4, which had been isolated from a U2OS T-REx cell line (*Figure 1E*). Thus, STIL and PLK4 form a stable complex in vivo.

## The STIL-CC motif is necessary for PLK4 binding and centriole duplication

To map the region of STIL required for binding to PLK4, we cloned truncated versions of the STIL protein: an N-terminal (STIL N-ter., residues 1–440), middle (STIL-MD, residues 441–880) and C-terminal (STIL C-ter., residues 881–1287) part, and subjected these fragments to co-immunoprecipitation experiments with myc-tagged PLK4-ND (*Figure 2A,B*). The PLK4-ND point mutant exhibits enhanced stabilization (*Guderian et al., 2010*) and thus facilitates the visualization of STIL binding (*Figure 1—figure supplement 2B*). We found that the N- and C-terminus of STIL did not bind PLK4-ND, whereas the middle part displayed efficient PLK4 binding. Accordingly, two STIL truncations

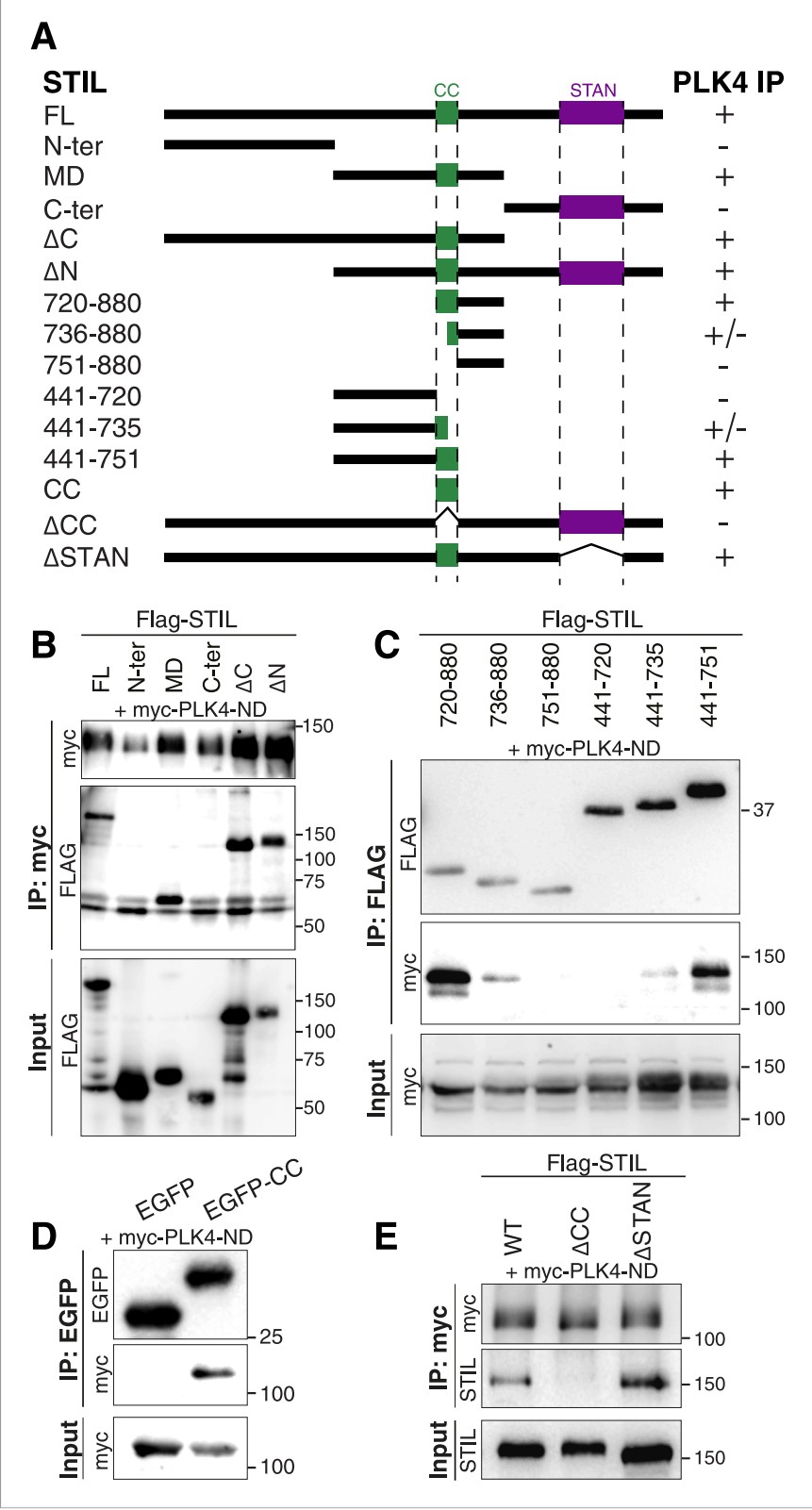

**Figure 2**. The STIL-CC motif binds to PLK4. (**A**) Schematic illustration of STIL constructs used to map the PLK4-binding region in STIL. On the right, the relative strengths of the interactions as determined by co-immunoprecipitation experiments are indicated (+, strong; ±, weak; -, not detected). (**B–E**) Western blot analysis of co-immunoprecipitation experiments from HEK293T cells co-expressing STIL fragments or STIL-ΔCC/ΔSTAN

*Figure 2. continued on next page*

*Figure 2. Continued*

mutants and myc-PLK4-ND. Cells were transfected for 36 hr with the indicated plasmids and whole cell lysates were used for co-immunoprecipitation experiments with anti-myc, anti-FLAG or anti-EGFP antibodies. Antibodies for Western blot detection are indicated.

containing the middle part but lacking either the N- or C-terminus (STIL-ΔN, residues 441–1287; STIL-ΔC, residues 1–880), strongly bound to PLK4-ND (*Figure 2A,B*).

The middle region of STIL contains a predicted CC motif (residues 720–751) (*Stevens et al., 2010*). To test the involvement of STIL-CC in PLK4 binding, we further truncated the STIL-MD and analyzed the interaction with PLK4-ND (*Figure 2A,C*). As long as the CC motif was intact, immunoprecipitation of PLK4-ND was not affected. However, truncating or removing the CC motif severely disrupted PLK4 binding, indicating that PLK4 interacts with the STIL-CC region. This finding was further confirmed by the observation that an EGFP-tagged version of STIL-CC efficiently pulled down PLK4 (*Figure 2D*), suggesting that STIL-CC alone is sufficient to bind PLK4. Accordingly, a mutant of STIL lacking the CC motif (STIL-ΔCC) did not interact with PLK4, whereas removal of another conserved region in STIL, the STAN domain (STIL-ΔSTAN), had no impact on the interaction with the kinase (*Figure 2A,E*). Therefore, the CC domain is both necessary and sufficient for STIL binding to PLK4.

Having established the importance of the CC motif for the PLK4/STIL interaction, we next tested the requirement of this motif for STIL functionality in centriole reduplication. Therefore, we transiently overexpressed STIL-ΔCC in U2OS cells and monitored centriole numbers using immunofluorescence microscopy (*Figure 3A,B*). Overexpression of wild-type STIL (STIL-WT) caused centriole amplification in 45% of transfected cells, and roughly 20% of cells displayed 'flower-like' staining, the near-simultaneous formation of several daughter centrioles around one mother centriole (*Figure 3A,B*). In cells overexpressing STIL-ΔCC only background levels of centriole amplification could be detected (*Figure 3B*). Moreover, we observed a similar reduction in centriole amplification with the STIL-ΔSTAN mutant, in line with the requirement for the STAN domain in centriole duplication (*Vulprecht et al., 2012*). Importantly, deletion of the STAN domain had only little impact on centriolar association of STIL, whereas removal of the CC motif strongly impaired the localization of STIL to centrioles (*Ohta et al., 2014*) (*Figure 3C*). Thus, together with the observation that PLK4 depletion leads to loss of STIL from centrioles (*Figure 1B*), these results indicate that PLK4 directly recruits STIL to the site of centriole formation. We next asked whether STIL overexpression has an impact on the localization of PLK4 to centrioles (*Figure 3C*). To this end, we overexpressed EGFP tagged STIL in U2OS cells and stained for endogenous PLK4 (*Figure 3D–G*). Under these conditions, STIL-WT triggered the near-simultaneous formation of several daughter centrioles and PLK4 formed a ring around preexisting centrioles, suggesting that STIL stabilizes centriolar PLK4 (*Figure 3D*). Overexpression of STIL-ΔCC had no effect on either centriole amplification or PLK4 localization (*Figure 3E*) and, most importantly, overexpression of STIL-ΔSTAN stabilized PLK4 at centrioles, even though this mutant did not cause formation of extra centrioles (*Figure 3F*). Therefore, we conclude that STIL can stabilize PLK4 at centrioles in a manner that is independent of centriole formation.

As STIL has been shown to self-associate (*Tang et al., 2011*), we also tested a possible involvement of the STIL-CC motif in self-interaction. We found that STIL-MD, and, more precisely, the CC motif, is indeed strictly required for STIL self-association, whereas the STAN domain is not (*Figure 3—figure supplement 1*). We conclude that the CC motif is critical for the function of STIL in centriole duplication through its role in PLK4 binding, STIL self-interaction, and STIL centriolar recruitment.

## STIL-CC directly binds PLK4-PB3 with nanomolar affinity

To determine which regions of PLK4 are involved in STIL binding, we generated a series of EGFP- and FLAG-tagged PLK4 fragments comprising either the N-terminal PLK4 part (residues 1–570), including the kinase domain (1–271) and the linker region L1 (265–570), or the C-terminal region containing the Polo-boxes (PB1/2 and PB3, residues 570–970; PB1/2, residues 570–820; L2-PB3, residues 814–970; PB3, residues 880–970) (*Figure 4A*). Co-expression of the EGFP-tagged PLK4 fragments with FLAG-tagged full-length STIL, followed by anti-EGFP co-immunoprecipitation, revealed that both, the N-terminal PLK4 fragment spanning 1–570 or all C-terminal fragments harboring PB3 were sufficient

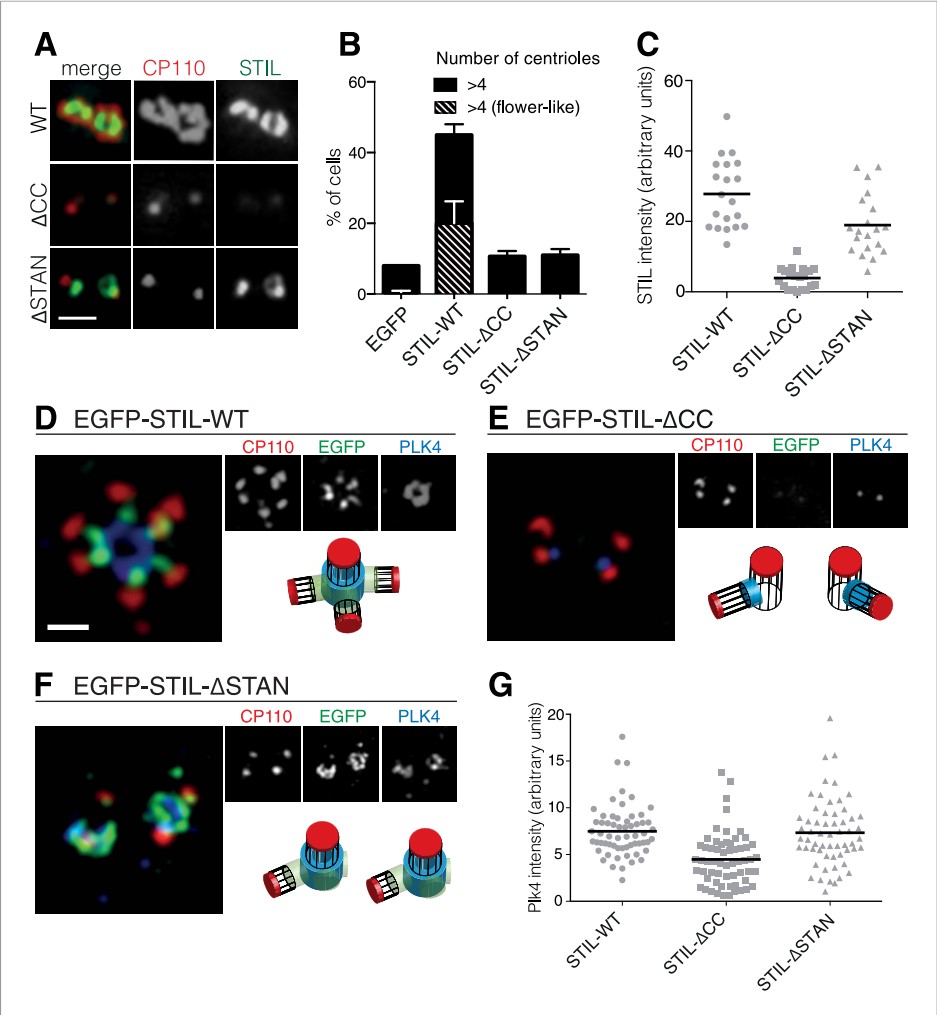

**Figure 3**. The STIL-CC motif is essential for centriole duplication. (**A**) Immunofluorescence microscopy of U2OS cells transfected with STIL-WT, STIL-ΔCC or STIL-ΔSTAN for 48 hr. Cells were fixed and stained with the indicated antibodies. Scale bar denotes 1 μm. (**B**) Quantification of centriole numbers in U2OS cells after overexpression of the indicated STIL plasmids (3 experiments, a total of 300 cells were analyzed for each condition). Error bars denote SD. (**C**) Scatter plot to illustrate STIL signal intensity at centrosomes, after overexpression of STIL-WT, STIL-ΔCC or STIL-ΔSTAN (20 centrosomes were analyzed for each condition). (**D**–**F**) 3D-SIM images of U2OS cells that have been transfected with EGFP-tagged STIL-WT, STIL-ΔCC and STIL-ΔSTAN and stained with the indicated antibodies. (**G**) Scatter plot to illustrate measured PLK4 signal intensities at centrosomes, after overexpression of STIL-WT/ΔCC or ΔSTAN (60 centrosomes were analyzed for each condition). Scale bar denotes 1 μm.

The following figure supplement is available for figure 3:

**Figure supplement 1**. The STIL-CC domain is essential for STIL oligomerization.

for the interaction with full-length STIL (*Figure 4B*). Moreover, the STIL-CC motif alone was sufficient to bind both the N-terminus (residues 1–570) and PB3 (residues 880–970) of PLK4 (*Figure 4C*). Within the N-terminal PLK4 part, the linker region (265–570) participated in the interaction, whereas the kinase domain itself (1–271) did not (*Figure 4D*). Attempts to further narrow down the interaction region in L1 were unsuccessful, indicating a non-linear folded binding region. To quantify the PLK4-PB3/STIL-CC interaction we determined the binding affinity of PLK4-PB3 and STIL-CC using isothermal titration calorimetry. For this purpose, PB3 (residues 884–970) was recombinantly expressed in *Escherichia coli* and purified, and a synthetic peptide corresponding to STIL-CC was subsequently titrated into a solution of PLK4-PB3. The integrated raw data are well fitted by a one-site

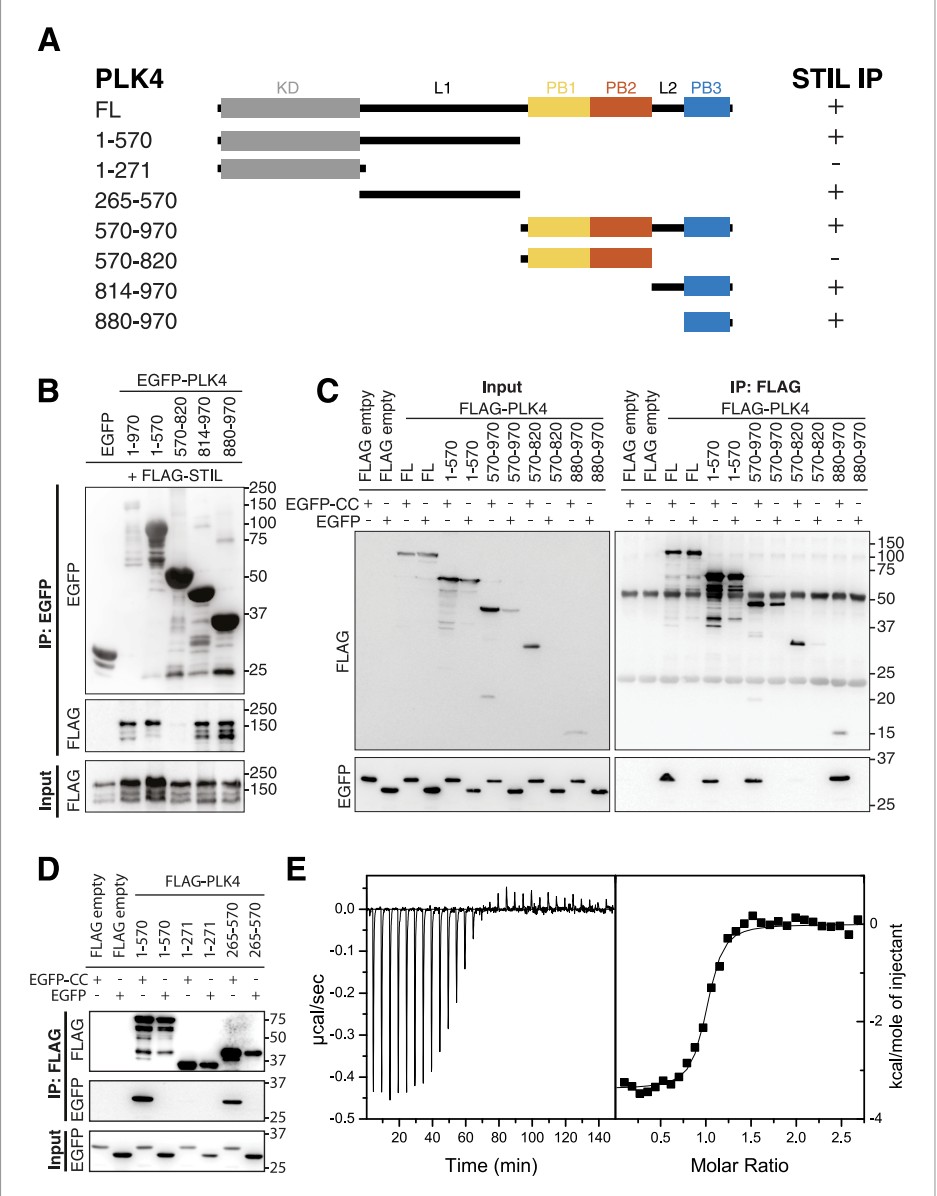

**Figure 4**. PB3 of PLK4 directly interacts with STIL-CC. (**A**) Schematic illustration of PLK4 fragments used to map the STIL-CC binding site. Kinase domain (KD), grey; PB1, yellow; PB2, orange; PB3, blue. The relative strengths of the interactions are indicated (+, strong; -, not detected). (**B**–**D**) Western blots of co-immunoprecipitation experiments using HEK293T cells co-transfected with plasmids expressing PLK4 fragments and FLAG-STIL (**B**) or EGFP-STIL-CC (**C** and **D**). Antibodies used for Western blot detection are indicated. (**E**) Isothermal titration calorimetry of STIL-CC into a solution of PLK4-PB3. Left panel: Direct measurement of the Gibbs energy associated with STIL-CC binding to PLK4-PB3. Right panel: integrated and fitted raw data using a one-site binding model.

binding model demonstrating a direct interaction between PB3 and STIL-CC with a dissociation constant $K_d$ of 280 ± 60 nM and an equimolar stoichiometry (*Figure 4E*).

In summary, we find that the STIL-CC motif directly interacts with PLK4-PB3 and that binding occurs with nanomolar affinity. Moreover, we show that STIL-CC additionally interacts with the N-terminus of PLK4, but not the PB1/2 domain. Importantly, our findings thus identify a novel binding mode for PLK4, since the previously known PLK4 binding partners Cep152 and Cep192 interact exclusively with the PB1/2 domain (*Cizmecioglu et al., 2010*; *Dzhindzhev et al., 2010*; *Hatch et al., 2010*; *Kim et al., 2013*;

*Sonnen et al., 2013*). As STIL is the first protein known to bind PLK4-PB3, we next characterized the structural basis of the PLK4-PB3/STIL-CC interaction.

## Structure determination and dynamics of PLK4-PB3 and its STIL-CC complex

NMR diffusion experiments confirmed that both free and STIL-bound PLK4-PB3 are monomeric in solution (*Supplementary file 1*). The structure of human PLK4-PB3 (residues 884–970) was determined by solution NMR spectroscopy (*Figure 5A*; *Table 1*). No crystals diffracting to high resolution were obtained for this construct, despite extensive screening of conditions. In contrast, plate-like crystals of the PLK4-PB3/STIL-CC complex diffracting to 2.6 Å resolution were obtained using seeding. The structure was solved by molecular replacement, and refined to $R_{work/free}$ of 0.22/0.25, respectively (*Table 1*) with one monomeric PLK4-PB3/STIL-CC per asymmetric unit (*Figure 5B*). The NMR structure of PLK4-PB3 in solution and the crystal structure of its STIL-CC complex display a conserved overall fold (*Figure 5C*) comprising a six-stranded antiparallel β-sheet (β1–β6) and a C-terminal α-helix (α1), which packs against the β-sheet and contacts residues from all six strands. The α-helical STIL-CC is bound in a hydrophobic cleft formed by both the β-sheet and the α1 helix of PLK4-PB3.

To further characterize differences in structure and dynamics of PLK4-PB3 seen upon STIL-CC binding in aqueous solution, free and STIL-CC-bound PLK4-PB3 were subjected to 2D [$^{15}$N,$^{1}$H]-TROSY experiments to reveal chemical shift perturbations and to $^{15}$N-{$^{1}$H}NOE measurements to characterize backbone dynamics on the ps- to ns-timescale (*Figure 5—figure supplements 1, 2*). Chemical shift perturbations are observed throughout most of the PB3 backbone and comprise direct STIL-CC interactions and perpetuated structural changes throughout PLK4-PB3. The most significant changes locate however around residues C954 and L955 on helix α1, where a slight kink is formed in apo

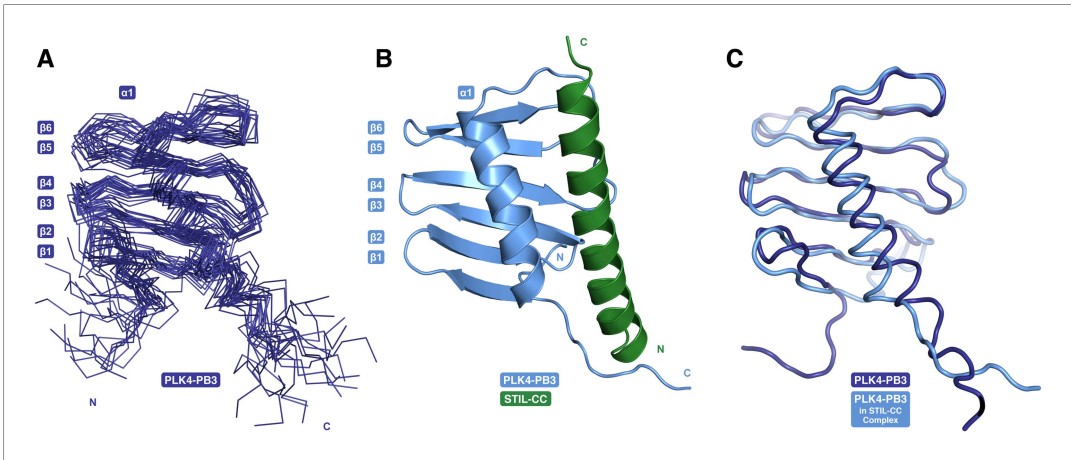

**Figure 5**. PB3 adopts a canonical Polo-box fold. (**A**) Ensemble of 20 NMR conformers with the lowest target function of free PLK4-PB3 (dark blue) (**B**) the X-ray structure of the PLK4-PB3/STIL-CC complex (light blue/green) and (**C**) comparison of the free PLK4-PB3 (dark blue) to the PLK4-PB3 (light blue) in complex with STIL-CC by structural superposition (STIL-CC not shown for clarity).

The following figure supplements are available for figure 5:

**Figure supplement 1**. Binding of STIL-CC to PLK4-PB3.

**Figure supplement 2**. Backbone dynamics of PLK4-PB3.

**Figure supplement 3**. Secondary structure elements of PLK4-PB3 in solution.

**Figure supplement 4**. PB3 adopts a canonical Polo-box fold.

**Table 1**. NMR and X-ray data collection and refinement statistics

| | PLK4-PB3 |
|---|---|
| **NMR distance and dihedral constraints** | |
| Distance constraints | |
| Total NOE | 399 |
| Intra-residue | 49 |
| Inter-residue | 350 |
| Sequential ($|i − j| = 1$) | 109 |
| Medium-range ($|i − j| < 4$) | 100 |
| Long-range ($|i − j| > 5$) | 141 |
| Hydrogen bonds | 16 |
| Total dihedral angle restraints | 122 |
| $\phi$ | 61 |
| $\psi$ | 61 |
| **Structure statistics** | |
| Violations (mean and s.d.) | |
| Distance constraints (Å) | 0.0158 ± 0.0028 |
| Dihedral angle constraints (°) | 1.686 ± 0.147 |
| Max. dihedral angle violation (°) | 8.40 |
| Max. distance constraint violation (Å) | 0.278 |
| Deviations from idealized geometry | |
| Bond lengths (Å) | 0.004 ± 0.000 |
| Bond angles (°) | 0.483 ± 0.017 |
| Impropers (°) | 1.391 ± 0.102 |
| Average pairwise r.m.s. deviation* (Å) | |
| Heavy† | 1.60 ± 0.21 |
| Backbone† | 1.09 ± 0.25 |

| | PLK4-PB3/STIL-CC |
|---|---|
| **X-ray data collection** | |
| Space group | C222$_1$ |
| Cell dimensions | |
| $a$, $b$, $c$ (Å) | 86.6, 136.3, 33.4 |
| $\alpha$, $\beta$, $\gamma$ (°) | 90.0, 90.0, 90.0 |
| Resolution (Å) | 68.1–2.60 (2.60–2.76) |
| $R_{merge}$ | 0.15 (2.37) |
| CC$_{1/2}$ outer shell | 0.63 |
| $I/\sigma I$ | 10.3 (0.9) |
| Completeness (%) | 99.7 (99) |
| Redundancy | 6.4 (6.6) |
| **Refinement** | |
| Resolution (Å) | 20.80–2.60 |
| No. reflections | 6409 |
| $R_{work}/R_{free}$ | 0.22/0.25 |

*Table 1. Continued on next page*

PLK4-PB3. Notably, the secondary chemical shifts of these residues are substantially increased upon binding STIL-CC, suggesting a stabilization of helical conformation (*Figure 5—figure supplement 3*). STIL-CC binding leads to an increase in averaged heteronuclear NOEs for β-strands and the α1 helix, suggesting a general reduction of fast backbone motions of PLK4-PB3 upon STIL-CC binding. Overall, the structural data reveal two core differences in PLK4-PB3 induced by STIL-CC binding: first, strand β1 is N-terminally extended by three residues to the range 888–893, resulting in a shortening of the unstructured N-terminal region in the STIL-CC complex. Second, helix α1 slightly changes its orientation and is stabilized by STIL-CC binding (*Figure 5C*).

## PLK4-PB3 adopts a canonical Polo-box fold

The structure of the human PLK4-PB3 resembles the canonical structures of related Polo-boxes (*Figure 5—figure supplement 4A*). It aligns structurally well with both Polo-boxes of the PLK1-PBD (*Cheng et al., 2003*; *Elia et al., 2003b*). PLK1-PB2 is a close structural homologue (1.3 Å rmsd, 79 Cα), with minor differences only in the linker to the α-helix. PLK1-PB1 is slightly more divergent, in that the C-terminal end of the α1-helix is bent towards the region, where STIL-CC is bound in PLK4-PB3. PLK4-PB3 also aligns with its two companion Polo-boxes of the PLK4-PB1/PB2 domain (*Slevin et al., 2012*; *Park et al., 2014*). Structural divergence to PLK4-PB1 occurs in the β-hairpin region between β3-β4, which gives PLK4-PB1 its unique winged structure, as well as in the linker between β5-β6. A major distinction between PLK4-PB3 and PLK4-PB2 is the length of helix α1, which extends beyond the β-sheet in PLK4-PB2.

PB3 of human PLK4 is closely related to those of its murine ortholog SAK (97% sequence identity [*Sievers et al., 2011*]). Based on the high degree of sequence identity one would expect a highly similar structure for this protein. However, the structure of the human PLK4-PB3 determined here diverges drastically from the structure of murine SAK-PB3, that was crystallized as a domain swapped dimer (*Leung et al., 2002*) (*Figure 5—figure supplement 4B*). In SAK-PB3, the β-sheet is formed by strands β2, β3 and β4 from one monomer and strands β5 and β6 from the second (numbering according to human PLK4-PB3) and the α1 helix is shortened compared to human PLK4-PB3. The sequence

Table 1. Continued

| | PLK4-PB3/STIL-CC |
|---|---|
| No. atoms | |
| Protein | 1829 |
| Water | 15 |
| B-factors | |
| Protein | 89.7 |
| Water | 78.6 |
| R.m.s. deviations | |
| Bond lengths (Å) | 0.01 |
| Bond angles (°) | 1.15 |

*Pairwise r.m.s. deviation was calculated among 20 refined structures.
†Statistics applied for ordered regions (residues 890–961).

region corresponding to the C-terminal half of this helix is swapped between monomers in SAK-PB3 and transformed into a β-strand, which occupies the position of strand β1 in human PLK4-PB3. The best explanation for this divergence might be the existence of an equilibrium between a monomeric and a domain-swapped form of SAK-PB3, the latter of which may be a lowly populated species that is not occurring in the full-length protein in vivo. Either the chemical conditions of crystallization shifted this equilibrium towards the non-native domain-swapped form or crystallization occurred selectively for the domain-swapped state. Nevertheless, our consistent results from solution and crystal structure determination strongly indicate that the canonical Polo-box fold is a relevant physiological state of the PLK4-PB3 domain.

## STIL-CC binding to PLK4-PB3 mimics CC interactions

PLK4-PB3 and STIL-CC interact along the entire STIL-helix and form a substantial interface of 934 Å² buried surface area (*Krissinel and Henrick, 2007*). Two regions on PLK4-PB3 contribute to the predominantly hydrophobic binding interface: First, the surface of the PLK4-PB3 β-sheet around residues V907, L917, V919, I926 and Y928 (*Figure 6—figure supplement 1*), and second the α1 helix (I948, L952, L955 and L959) and the linker (L944) leading into it (*Figure 6A*). Key interacting residues on STIL-CC are leucine and isoleucine residues (L733, L736, I740, L743, L744) pointing towards the hydrophobic surface of PLK4-PB3. Additional interactions are provided by backbone–backbone hydrogen bonds between PB3$_{G922}$-STIL$_{Q739}$ and PB3$_{K943}$-STIL$_{M750}$ (*Figure 6—figure supplement 1*), respectively. The orientation of the two helices and their hydrophobic interactions is mainly mediated by leucine residues and resembles a leucine zipper interaction, consistent with the predicted CC propensity of STIL-CC (*Stevens et al., 2010*).

Polo-box domains are crucial mediators of the interaction of Polo-like kinases with their targets and have been demonstrated to interact with irregular substrate peptides and phosphopeptides. PLK1, for example, binds phosphopeptides containing a consensus Ser-[pSer/pThr]-[Pro/X] motif (*Elia et al., 2003a*) through a cleft within its PBD (comprising PB1 and PB2) (*Cheng et al., 2003*; *Elia et al., 2003b*; *Sledz et al., 2011*) and a neighboring binding site on PB1 is used for phospho-independent recognition of a Map205 peptide (*Xu et al., 2013*) (*Figure 6—figure supplement 2A*). The PLK4-PB1/2 domain has recently been shown to bind either two Cep192- or one Cep152-derived peptide in a mutually exclusive manner using large interaction interfaces extending along PB1 and PB2 (*Park et al., 2014*). PLK4-PB3 reveals a novel binding mode by interacting with the helical STIL-CC region in a leucine-zipper-style via the α1 helix with further hydrophobic contacts to the central β-sheet. Remarkably, this external target interaction of PLK4-PB3 closely resembles an intramolecular interaction observed in PLK1: there, the internal Polo-cap (Pc), an N-terminal extension of PLK1-PB1 which comprises an α-helix and a linker to PB1, directly binds to PLK1-PB2 and thereby determines the relative orientation of PB1 and PB2 (*Elia et al., 2003b*). The 17-residue α-helix of the Pc is shorter compared to the 31-residue STIL-CC, but forms a leucine-zipper with the α1-helix of PLK1-PB2, very much like STIL-CC with the α1-helix of PB3 in PLK4 (*Figure 6—figure supplement 2B*).

## Mutations demonstrate the relevance of PLK4/STIL-CC interactions in vivo

Guided by the structure of the PLK4-PB3/STIL-CC complex, we designed seven mutants of STIL to functionally evaluate the role of the PLK4-PB3 interaction with STIL (mutants M1 to M7): in the first mutant (M1) L733 and L743 were replaced by two large residues (L733Y, L743Y). In the second mutant (M2) as well as in the mutants M4-7 we replaced hydrophobic residues (at positions a and d of the CC heptads) with charged amino acids. In mutant M3 we exchanged Q737E and Q739K in order to

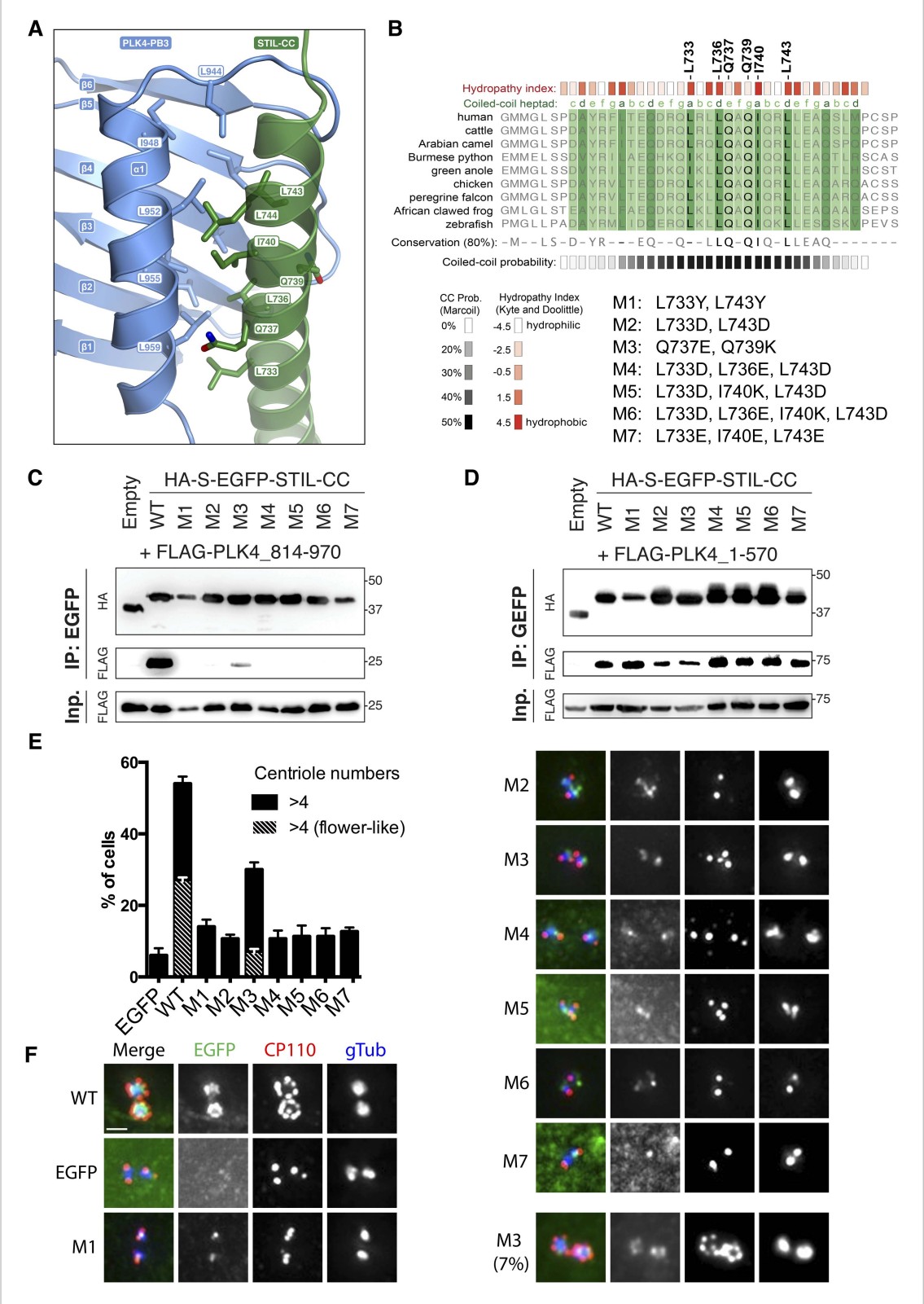

Figure 6. Analysis of structure-based STIL-CC mutants. (A) Close-up view of the STIL-CC/PLK4-PB3 binding interface. Key contributing residues to the hydrophobic interaction between the PLK4-PB3 (light blue) α-helix and the STIL-CC (green) α-helix are indicated. (B) Multiple sequence alignment (ClustalW) of the STIL-CC domain. Residues that directly participate in the PLK4-PB3 interaction are marked on top. Hydropathy index values (according to Kyte-Doolittle) for each amino acid are depicted in red. CC probability values (according to MARCOIL prediction) for each amino acid are depicted in

*Figure 6. continued on next page*

*Figure 6. Continued*

black below the alignment. The position of each amino acid in the predicted heptad repeat (labelled a-g, whereas a and d are hydrophobic positions) is shown in green on top of the alignment. (**C, D**) Control, WT or mutant versions (M1-7) of HA-S-EGFP-tagged STIL-CC were co-transfected with either PLK4-L2-PB3 (814–970) (**C**) or N-terminal PLK4 (1–570) (**D**) in HEK293T cells. EGFP-immunoprecipitations were performed and analysed by Western blotting with the indicated antibodies. (**E, F**) To assess centriole amplification, EGFP-tagged WT and mutants of full-length STIL (M1-7) were overexpressed in U2OS cells (48 hr). EGFP was used as control. (**E**) Quantification of transfected cells with more than 4 centrioles (n = 3, 50 cells each). (**F**) Immunofluorescence images of U2OS cells after overexpression of EGFP-tagged STIL-WT or mutants M1-7 (48 hr). Centrioles were visualized by staining with antibodies against CP110 and γ-Tubulin (gTub) was used as marker for centrosomes. Scale bar: 1 μm.

The following figure supplements are available for figure 6:

**Figure supplement 1**. STIL–CC binding to PLK4-PB3 mimics coiled-coil interactions.

**Figure supplement 2**. STIL–CC binding to PLK4-PB3 resembles an intramolecular interaction of PB2 and Pc in PLK1.

interfere with hydrogen bond formation (STIL$_{Q739}$-PB3$_{K943}$) (*Figure 6B*). To test the impact of these mutations on the binding affinity to PLK4, we co-transfected HA-S-EGFP-tagged mutants of STIL-CC with either FLAG-tagged PLK4-L2-PB3 (residues 814–970) or N-terminal PLK4 (residues 1–570). The corresponding EGFP-immunoprecipitation experiments revealed that all mutants except for M3 completely lost binding to PB3 (*Figure 6C*). Thus, the hydrophobic residues L733 and L743 as well as L736 and I740 are indeed essential for the interaction. Disruption of a hydrogen bond at position Q739 in M3 severely diminished the interaction with PB3 but still allowed for residual binding. Interestingly, all mutants maintained binding to the N-terminal PLK4 fragment spanning residues 1–570 (*Figure 6D*). This indicates that STIL-CC binds to the two PLK4 regions in different ways and hence may associate with the two regions simultaneously.

In parallel to the above binding studies, we also analysed the ability of the STIL mutants to cause centriole amplification. We overexpressed EGFP-tagged full-length WT or mutant STIL proteins in U2OS cells and scored for cells with more than four centrioles. STIL-WT produced centriole amplification in 54% of cells, half of them displaying a flower-like centriole arrangement. In contrast, the mutants M1-2 and M4-7 caused centriole amplification in only 11–14% of cells, comparable to centriole amplification in EGFP-transfected control cells (*Figure 6E,F*). Interestingly, in the case of M3, 30% of cells produced amplified centrioles (7% with a flower-like centriole arrangement) (*Figure 6E, F*), in line with the observed residual binding capacity to PB3 (*Figure 6C*).

## Discussion

Polo-like kinases are a family of kinases with key regulatory roles in cell cycle progression, mitosis, cytokinesis and centriole duplication, with all human genes thought to have arisen by gene duplication from an ancestral PLK1-like gene (*Zitouni et al., 2014*). PLK4, the master regulator of centriole duplication, is the most distant member of the family. It is distinguished from all other PLKs by the presence of a third C-terminally located Polo-box, PB3, in addition to the two central and closely linked Polo-boxes PB1 and PB2 (*Zitouni et al., 2014*). PB1 and PB2 provide a dimerization platform to regulate PLK4 *trans*-autophosphorylation (*Klebba et al., 2015*) and mediate recognition of crucial interaction partners (*Slevin et al., 2012*; *Park et al., 2014*; *Zitouni et al., 2014*). However, no interaction partner of PLK4-PB3 had previously been identified.

Here, we demonstrate that monomeric PB3 of human PLK4 adopts a canonical Polo-box-fold and directly interacts with nanomolar affinity with the central STIL-CC region, STIL-CC. The binding mode of the PLK4-PB3/STIL-CC interaction is completely different from all previously described target interactions of Polo-boxes, but resembles an intramolecular interaction of PB2 with the Pc-helix in PLK1.

We show that the interaction of STIL with PLK4 in vivo regulates centriole biogenesis, confirming and extending recent reports that PLK4-mediated STIL phosphorylation is crucial for SAS-6 recruitment and for triggering centriole duplication (*Dzhindzhev et al., 2014*; *Ohta et al., 2014*; *Kratz et al., 2015*). We further demonstrate that the STIL-CC region is necessary and sufficient to mediate the STIL-PLK4 interaction and map the binding of STIL-CC on PLK4 to two distinct domains,

first to PB3, and second, to the L1 linker between the kinase and PB1/2 domain. Previous studies indicated that either the N-terminal kinase and L1 (*Kratz et al., 2015*) or, alternatively, the PB1/2 domain (*Ohta et al., 2014*) of PLK4 are required for STIL interaction, but had failed to reveal an involvement of PB3. However our in vivo, quantitative biophysical and structural data firmly establish a PLK4-PB3/STIL-CC interaction.

The binding of STIL-CC to PB3 resembles a zipper-type CC interaction based on an amphiphilic helix in PB3. However, the L1 linker region is predicted to not contain any Polo-box folds or other regions favouring CC interactions. This suggests different interaction modes between STIL-CC and PB3 or L1, respectively, and raises the exciting prospect that STIL-CC could regulate internal interactions between PB3 and the L1 linker region. As further discussed below, this may constitute an important mechanism for regulating PLK4 activity. Furthermore, our data clearly demonstrate a dual role for the STIL-CC region: first, it is an interaction partner for PLK4-PB3 in its monomeric form, and, second, it is involved in STIL-self interactions, presumably via tetramerization (*Cottee et al., 2015*). Mutations in the CC motif clearly abolish the interaction with PLK4-PB3. However, the same hydrophobic residues of STIL-CC are also predicted to be essential for STIL self-interaction and it is thus likely that these two processes are coupled. Therefore, the inability of STIL-CC mutations to support centriole amplification can be due to either compromised PLK4-PB3 binding, lack of STIL self association or due to failure in both processes. Furthermore, PB3-binding to STIL-CC is expected to affect the self-association of STIL with potential consequence for protein–protein interactions downstream of STIL.

Recent studies have revealed a regulatory role for PB3 in the activation of SAK/PLK4 in *Drosophila* (*Klebba et al., 2015*). Specifically, it was suggested that PB3, autophosphorylation of L1, and potentially further yet unidentified protein partners are important for relief of kinase auto-inhibition after SAK/PLK4 homodimerization. Our data on human PLK4 suggest to attribute a key role in such regulatory mechanism to STIL: first, STIL is the only identified interaction partner of PB3 with a direct influence on PB3 structure and dynamics; second, through its CC domain STIL is also able to interact with purported regulatory regions within L1 of human PLK4. These features make STIL a prime candidate for the role of an external factor regulating relief of PLK4 auto-inhibition in time and space (*Figure 7*, upper panel).

Although some mechanistic aspects of centriole biogenesis are likely to differ between species (*Kim et al., 2014*) it is interesting to consider our structural data on the interaction between PLK4 and STIL-CC in light of recent insights into the first steps of centriole formation. Specifically, our data supports a model where STIL binds to PLK4 that has been recruited to centrioles through interactions with CEP192 and/or Cep152 (*Figure 7*, lower panel), either around the centriolar ring or in a localized dot-like pattern. Additional contributions to stable localization of STIL to centrioles may arise from its STAN-motif or C-terminal region, as a STIL C-terminal truncation (amino acids 1–1060) that contains the CC domain, interfered with correct localization to the centrioles (*Vulprecht et al., 2012*; *Arquint and Nigg, 2014*). Similarly, the isolated CC domain of Ana2 is not sufficient for centriolar targeting in *Drosophila* embryos (*Cottee et al., 2015*). Therefore, further interactions, presumably between the STAN-motif or the C-terminal region of STIL and SAS-6, are likely required to stably integrate STIL into the centriolar cartwheel structure.

Once in a complex with PLK4, STIL is phosphorylated, triggering the recruitment of SAS-6 in preparation for cartwheel formation (*Dzhindzhev et al., 2014*; *Ohta et al., 2014*; *Moyer et al., 2015*). In parallel, phosphorylation may affect STIL oligomerization and association with the walls of centrioles. In view of most recent studies on PLK4 activation, our data additionally indicate that STIL-CC relieves the auto-inhibition of PLK4 (*Klebba et al., 2015*; *Moyer et al., 2015*). The resulting PLK4 *trans*-autophosphorylation is predicted to cause recruitment of βTrCP-SCF, followed by ubiquitin-dependent proteasomal degradation of activated PLK4 (*Guderian et al., 2010*; *Holland et al., 2010*). In line with this idea, depletion of STIL leads to remarkable increase of PLK4 levels (*Figure 1C* and *Figure 1—figure supplement 1*), suggesting that PLK4 is accumulating in an inactive and hence stabilized conformation in the absence of STIL. Our data also indicate that STIL protects activated PLK4 from degradation (*Figure 3D–G*) and (*Ohta et al., 2014*)), possibly through binding to the PLK4-L1 linker region, which might shield the phosphorylated degradation motif (DSGHAT, residues 284–289) from recognition by βTrCP-SCF. Overall, these mechanisms provide a possible explanation for the concentration of PLK4 at the site of centriole formation (*Figure 7*, lower panel).

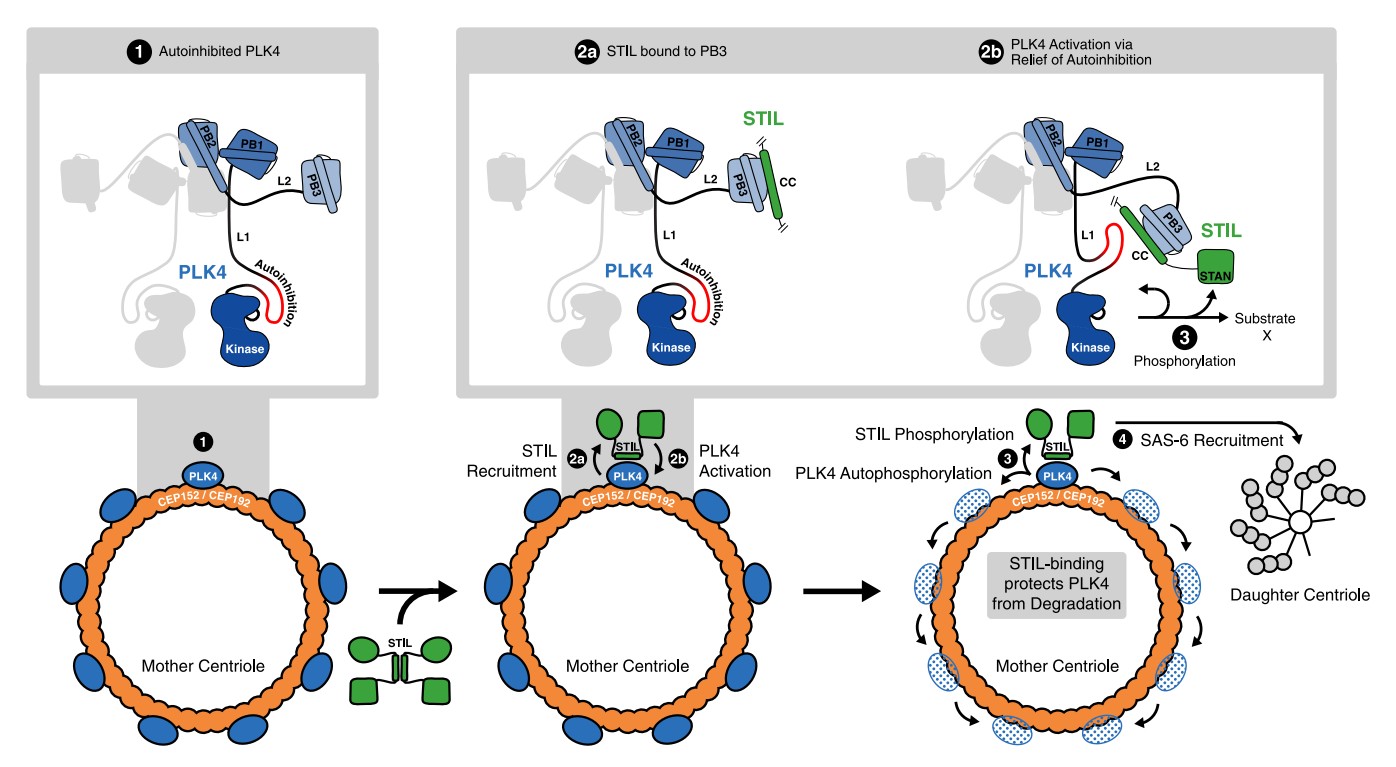

**Figure 7**. STIL binding to PLK4 regulates centriole duplication. Hypothetical mechanism for STIL-mediated PLK4 activation: (1) PLK4 is bound to the mother centriole. It is intrinsically inactive, likely due to an autoinhibition by linker L1. (2a) STIL binds to PLK4 that has been recruited to centrioles through interactions with CEP192 and/or Cep152. (2b) STIL binding relieves the autoinhibition of PLK4, thus activating PLK4. (3) Activated PLK4 phosphorylates STIL in the STAN motif, which induces SAS-6 recruitment and daughter centriole biogenesis (4). Activated PLK4 also phosphorylates neighboring PLK4s in the degradation motif, triggering their degradation. At the site of cartwheel formation, the STIL-bound PLK4 is protected against degradation.

In summary, we have identified and structurally characterized an interaction between STIL and PLK4, two key centriole duplication factors. We show that the interaction is mediated via the STIL-CC domain and is crucial for centriole biogenesis. Importantly, STIL-CC is the first *bona fide* interaction partner of PLK4-PB3. The interaction of STIL-CC with PLK4-PB3 and a second region within the PLK4 L1 linker likely results in PLK4 activation and STIL phosphorylation by PLK4. These novel insights into the interaction and crosstalk of two key factors in centriole biogenesis provide a new perspective for further work on the critical step of condensing a PLK4 ring to a spot in localizing the initial position of daughter centriole growth.

## Materials and methods

### Experimental procedures

#### Antibodies

Antibodies against Cep152 (rabbit) (*Sonnen et al., 2012*), Centrin-3 (rabbit) (*Thein et al., 2007*) CP110 (rabbit) (*Schmidt et al., 2009*), CP110 (mouse) (*Arquint and Nigg, 2014*), myc (9E10, mouse) (*Evan et al., 1985*), PLK4 (mouse) (*Guderian et al., 2010*), SAS-6 (mouse) (*Kleylein-Sohn et al., 2007*), and STIL (rabbit) (*Arquint et al., 2012*) were described previously. Monoclonal anti-FLAG M2 (Sigma–Aldrich, St. Louis, MI), rabbit polyclonal anti-GFP (Abcam, Cambridge, UK), monoclonal anti-γ-Tubulin TU-30 (Abcam), polyclonal anti-STIL (Abcam) and monoclonal anti-PLK4 MABC544, clone 6H5 (Merck Millipore, Billerica, MA) were purchased from the respective manufacturers. Alexa-555-, Alexa-488-, Alexa-647-labeled secondary anti-mouse and anti-rabbit antibodies were purchased from life technologies (Carlsbad, CA). Whenever appropriate, antibodies were directly labeled with Alexa-555, Alexa-488 and Alexa-647 fluorophores, using the corresponding Antibody Labeling Kits (life technologies).

## Plasmids

Cloning of myc-PLK4 full-length WT (*Habedanck et al., 2005*), myc-PLK4 full-length ND (S285A/T289A) (*Guderian et al., 2010*) and FLAG-STIL (*Arquint et al., 2012*) was described previously. pgLAP1 was a gift from Peter Jackson (Addgene plasmid #19702). All other plasmids were generated via the Gateway technology (life technologies) using the pENTR/D-TOPO entry vector (life technologies) and pcDNA3.1-based destination vectors containing the appropriate N-terminal tags. Mutations in STIL were generated with the QuikChange II XL Site-Directed Mutagenesis Kit (Agilent, Santa Clara, CA) according to the manufacturer's instructions.

## Kinase assays

In vitro kinase assays using recombinant GST-PLK4 1–430 WT/D154A (cloned into pGEX-5X-2 [Thesis J Westendorf]) were carried out at 30°C in kinase buffer (50 mM HEPES pH 7.0, 100 mM NaCl, 10 mM MgCl2, 5% glycerol, 1 mM DTT). Reactions were stopped after 30 min by addition of Laemmli sample buffer. Samples were then analyzed by autoradiography and Western blotting.

## Cell culture and transfections

U2OS and HEK293T cells were grown at 37°C and 5% CO2 in Dulbecco's modified Eagle's medium (DMEM), supplemented with 10% heat-inactivated fetal calf serum (FCS, PAN Biotech, Aidenbach, Germany) and penicillin-streptomycin (100 µg/ml, Gibco-BRL, Karlsruhe, Germany). The U2OS cell line harboring the tetracycline-inducible myc-PLK4 transgene has been described (*Kleylein-Sohn et al., 2007*). The T-REx U2OS Flp-IN cell line harboring the tetracycline-inducible S-peptide-EGFP-tagged PLK4 fragment (residues 570–970) has been generated as described previously (*Arquint and Nigg, 2014*). Transgene expression was induced by the addition of 1 µg/ml tetracycline. PLK4, STIL and βTrCP were depleted using siRNA oligonucleotides described previously (*Guardavaccaro et al., 2003*; *Habedanck et al., 2005*; *Arquint et al., 2012*). Transfections of siRNA oligonucleotides or plasmids were performed using the transfection reagents Oligofectamin (Invitrogen) or TransIT-LT1 (Mirus Bio, Madison, WI), respectively, according to the manufacturer's instructions.

## Cell extracts, immunoprecipitation, Western blots and mass spectrometry

Cell lysis was performed as described previously (*Guderian et al., 2010*) using Tris lysis buffer (50 mM Tris–HCl, pH 7.4, 150 mM NaCl, 0.5% IgePal) supplemented with protease and phosphatase inhibitors. For immunoprecipitation experiments, cell extracts (2–5 mg total protein) were incubated for 2 hr at 4°C with appropriate beads (Anti-FLAG M2 Affinity Gel [Sigma–Aldrich], GFP-Trap agarose [ChromoTek, Planegg, Germany], Affi-Prep protein A matrix [Bio-Rad Laboratories, Hercules, CA] crosslinked with 9E10 anti-myc antibodies or S-protein Agarose beads [Merck Millipore]). Immunocomplexes were washed 4–6 times with wash buffer (50 mM Tris–HCl pH 7.4, 150–500 mM NaCl, 0.5–1% IgePal), eluted with Laemmli buffer and analyzed by Western blotting. For mass spectrometry analysis of proteins copurifying with the S-peptide-EGFP-PLK4 fragment (residues 570–970), transgene expression was induced for 24 hr with Tetracycline (1 µg/ml). Cell extracts were subjected to S-protein immunoprecipitations and the samples were processed for mass spectrometry as described previously (*Maze et al., 2014*).

## Immunofluorescence microscopy

Cells were fixed in methanol (5 min at −20°C) and processed for immunofluorescence microscopy as described previously (*Meraldi et al., 1999*). Stainings were analyzed using a DeltaVision microscope on a Nikon TE200 base (Applied Precision, Issaquah, WA), equipped with a Plan Apochromat 60× 1.42 and an APOPLAN 100 × 1.4 N.A. oil-immersion objective (Olympus, Tokio, Japan), and a CoolSNAP HQ2 camera (Photometrics, Tucson, AZ). Serial optical sections acquired 0.2 µm apart along the z-axis were processed and projected into one image using Softworx (Applied Precision). For quantifications of STIL and PLK4 levels, intensities were measured with ImageJ and background signal intensity was subtracted. Identical image acquisition and image processing settings were used whenever measurements were compared. 3D-SIM was performed on a DeltaVision OMX-Blaze microscope system as described previously (*Sonnen et al., 2013*).

## Expression and purification of PLK4-PB3 for structural studies

The human PLK4-PB3 (884–970) was cloned into pETG-30A (EMBL, Heidelberg, Germany) with a N-terminal thioredoxin and hexahistidine tag, followed by tobacco etch virus (TEV) protease cleavage site (Gateway Cloning, life technologies). The vector was transformed into BL21 DE3 cells. The

expression culture was induced with 0.5 mM IPTG at an $OD_{600nm}$ of 0.5 and then grown for 18 hr at 20°C. The protein was purified using High Affinity Nickel Charged Resin (Genscript, Piscataway, NY). The N-terminal tag was removed by incubation with tobacco etch virus protease and rebinding to High Affinity Nickel Charged Resin. The PLK4-PB3 was further purified by size exclusion chromatography using a Hiload 16/60 Superdex75 column (GE, Fairfield, CT). Based on this procedure, the following isotope-labeled samples were prepared for NMR experiments: [$U$-99%-$^2$H, $^{15}$N]-PLK4-PB3, [$U$-99%-$^2$H, $^{15}$N, $^{13}$C]-PLK4-PB3, [$U$-99%-$^2$H, $^{15}$N; 99%-$^1$H$^\delta$,$^{13}$C$^\delta$-IL;99%-$^1$H$^\gamma$,$^{13}$C$^\gamma$-V]-PLK4-PB3, and [$U$-99%-$^2$H, $^{15}$N,$^{13}$C; 99%-$^1$H$^\delta$-IL;99%-$^1$H$^\gamma$,$^{13}$C$^\gamma$-V]-PLK4-PB3. All isotope-labeled reagents were purchased from Sigma–Aldrich.

## Isothermal titration calorimetry

Isothermal titration calorimetry measurement employed a MicroCal VP-ITC Instrument (Malvern Instruments, Worcestershire, UK). Recombinant PB3 (35.5 µM) and synthetic STIL-CC Peptide (414 µM) were dialyzed against 50 mM Hepes pH 8.0, 150 mM NaCl and 1 mM TCEP. The experiment was conducted at 25°C. Titrations employed 30 STIL-CC injections (10 µl) each followed by a 300 s delay. The experimental data was fit with a one-site binding model.

## NMR spectroscopy and structure determination of PLK4-PB3

All protein samples were prepared in 20 mM MOPS pH 7, 30 mM NaCl and 2 mM TCEP in 5%/95% $D_2O/H_2O$ with a protein concentration between 0.6 and 1.2 mM. All NMR spectra were recorded at 20°C on Bruker Avance (Bruker, Billerica, MA) spectrometers equipped with cryogenic triple-resonance probes (field strengths 700, 800 and 900 MHz). For the sequence-specific backbone resonance assignment of [$U$-$^2$H, $^{15}$N, $^{13}$C]-labeled PLK4-PB3 with and without bound STIL, the following NMR experiments were recorded: 2D [$^{15}$N, $^1$H]-TROSY-HSQC (*Pervushin et al., 1997*), 3D TROSY-HNCA, 3D TROSY-HNCACB, 3D TROSY-HNCO and 3D TROSY-HNCACO (*Salzmann et al., 1998*). Chemical shift perturbations ($\Delta\delta$(HN)) of amide moieties were calculated as: $\Delta\delta(HN) = [((\delta H_{ref} - \delta H)^2 + ((\delta N_{ref} - \delta N)/5)^2)/2]^{0.5}$. For the sequence-specific side chain resonance assignment of [$U$-$^2$H, $^{15}$N, $^{13}$C-methyl-ILV]-labeled PB3, the following experiments were recorded: 3D (H)C(CC)-TOCSY-(CO)-$^{15}$N-HSQC and 3D H(C)(CC)-TOCSY-(CO)-$^{15}$N-HSQC with a TOCSY mixing time of 26.6 µs (*Grzesiek and Bax, 1993*). The NOE distance restraints for structural calculation were derived from 3D [$^1$H,$^1$H]-NOESY-$^{15}$N-TROSY (*Marion et al., 1989*; *Zuiderweg and Fesik, 1989*) and 3D [$^1$H,$^1$H]-NOESY-$^{13}$C-HMQC (*Zuiderweg and Fesik, 1989*) with a NOE mixing time of 100 ms. For determination of the dynamic properties of [$U$-$^2$H, $^{15}$N]-PLK4-PB3 with and without bound STIL, $^{15}$N{$^1$H}-NOE experiments were measured (*Zhu et al., 2000*). $^{15}$N-filtered BPP-LED-diffusion experiments were recorded to measure the lateral diffusion of [$U$-$^2$H, $^{15}$N]-PLK4-PB3, [$U$-$^2$H, $^{15}$N]-PLK4-PB3/STIL-CC and [$U$-$^2$H, $^{15}$N]-GB1 as a control (*Gronenborn et al., 1991*; *Chou et al., 2004*).

Data were processed using Prosa (*Guntert et al., 1992*) and analyzed with CARA (*Keller, 2004*). The backbone assignment was done manually using CARA. Structure calculations were performed using CYANA 3.96 (*Herrmann et al., 2002*), based on NOE restraints and dihedral angle restraints calculated by Talos+ (*Shen et al., 2009*). Hydrogen bonding restraints were added for residues in secondary structural elements that were unambiguously predicted by Talos+, and that showed the characteristic backbone NOE pattern for these secondary structural elements. A total of 100 structures were calculated by CYANA and the 20 conformers with the lowest target function were selected. A final water-refinement step of 15 ns was performed with CNS (*Brunger et al., 1998*) Ramachandran statistics were 90.7% in most favored regions, 7.2% in additionally allowed regions, 1.7% in generously allowed regions and 0.4% in disallowed regions. Structures were analyzed using Molmol (*Koradi et al., 1996*).

## Crystallization and structure determination of PLK4-PB3/STIL-CC

Synthetic STIL-CC peptide was incubated with recombinant PLK4-PB3 on ice for 1 hr in 20 mM HEPES-KOH pH 8.0, 150 mM NaCl and 2 mM TCEP in a 1.2-fold molar excess. The complex was concentrated to 16 mg ml$^{-1}$ and set up for crystallization using the sitting-drop vapour diffusion method with a 1:1 (vol/vol) ratio of the complex and precipitant solutions. Thin needles grew in precipitant solution containing 100 mM Hepes, pH 7.0, 20 mM $MgCl_2$ and 22% Poly(acrylic acid sodium salt) 5100. After extensive optimization and seeding plate-like crystals were grown in a drop consisting of 50% PLK4-PB3/STIL-CC complex solution, 33.3% precipitant solution (100 mM phosphate/citrate pH 4.2, 40% (vol/vol) Ethanol, 5% (wt/vol) PEG 1000) and 16.7% of seed stock solution. Crystals were flash frozen in liquid nitrogen using Perfluoropolyether (Hampton Research, Aliso Viejo, CA) as cryoprotectant.

X-ray diffraction data was collected at the beamline X06DA (PXIII) at the Swiss Light Source (Paul Scherrer Institute, Switzerland) using a PILATUS 2M detector. Crystals in the space group C222$_1$ diffracted to 2.6 Å resolution. The X-ray diffraction data was processed using XDS (*Kabsch, 2010*). The structure was solved by molecular replacement with Phaser (*McCoy et al., 2007*) using PLK1-PB1 (3P34, residues 147–331) as an initial search model. The asymmetric unit comprises one monomeric PLK4-PB3/STIL-CC complex. Model building and refinement was performed with Coot (*Emsley and Cowtan, 2004*) and PHENIX (*Adams et al., 2002*) to $R_{work/free}$ of 0.22/0.25, respectively.

### Accession numbers
Atomic coordinates for PLK4-PB3 have been deposited in the Protein Data Bank with accession code 2n19 (*Böhm et al., 2015a*). Sequence-specific resonance assignments have been submitted to the Biological Magnetic Resonance Data Bank under the following codes: PLK4-PB3, 25552 and holo PLK4-PB3 with bound STIL-CC, 26547 (*Böhm et al., 2015b, 2015c*). Atomic coordinates and structure factors for PLK4-PB3/STIL-CC have been deposited in the Protein Data Bank with accession code 4YYP (*Arquint et al., 2015*).

## Acknowledgements

Crystallographic experiments were performed on beamline x06da at the Swiss Light Source, Paul Scherrer Institut, Villigen, Switzerland. This work was supported by the Swiss National Science Foundation grant 310030B_149641 to EAN and by R'Equip 145023. CA was funded by a PhD fellowship from the Werner Siemens-Foundation (Zug, Switzerland). We thank Stephen C Blacklow (Harvard Medical School), Jeff Parvin (Ohio State University Medical Center), Peter Jackson (Genentech) and Ingrid Hoffmann (Deutsches Krebsforschungszentrum) for sharing cell lines and reagents. We further acknowledge outstanding technical assistance by Elena Nigg and support by the Imaging, Biophysics and Proteomics Core Facilities of the Biozentrum.

## Additional information

### Funding

| Funder | Grant reference | Author |
|---|---|---|
| Schweizerische Nationalfonds zur Förderung der Wissenschaftlichen Forschung | 310030B_149641 | Christian Arquint, Anna-Maria Gabryjonczyk, Erich A Nigg |
| Schweizerische Nationalfonds zur Förderung der Wissenschaftlichen Forschung | R'Equip 145023 | Timm Maier |
| Werner Siemens-Stiftung | | Christian Arquint |

The funders had no role in study design, data collection and interpretation, or the decision to submit the work for publication.

### Author contributions
CA, A-MG, Biochemical in vivo and in vitro characterization of STIL/PLK4 interaction, contribution in the preparation of the manuscript, Acquisition of data, Analysis and interpretation of data, Drafting or revising the article; SI, ITC experiments, expression and purification of samples for NMR and X-ray crystallography, determination of the crystal structure of PLK4-PB3/STIL-CC, contribution in the preparation of the manuscript, Acquisition of data, Analysis and interpretation of data, Drafting or revising the article; RB, NMR analyses and determination of the NMR solution structure of PLK4-PB3, contribution in the preparation of the manuscript, Acquisition of data, Analysis and interpretation of data, Drafting or revising the article; ES, Contribution to protein expression and purification and the preparation of the manuscript, Analysis and interpretation of data, Drafting or revising the article; SH, EAN, TM, Design of experiments, supervision of the work, data analysis and contribution in the preparation of the manuscript, Conception and design, Analysis and interpretation of data, Drafting or revising the article

## Additional files

### Supplementary file

- Supplementary file 1. Molecular mass determination by lateral diffusion measurements.

### Major datasets

The following datasets were generated:

| Author(s) | Year | Dataset title | Dataset ID and/or URL | Database, license, and accessibility information |
|---|---|---|---|---|
| Böhm R, Arquint C, Gabryjonczyk A, Imseng S, Sauer E, Hiller S, Nigg E, Maier T | 2015 | STIL binding to the Polo-box domain 3 of PLK4 regulates centriole duplication | http://www.rcsb.org/pdb/explore/explore.do?structureId=2N19 | Publicly available at the RCSB Protein Data Bank (accession no. 2N19). |
| Arquint C, Gabryjonczyk AM, Imseng S, Böhm R, Sauer E, Hiller S, Nigg EA, Maier T | 2015 | Crystal structure of human PLK4-PB3 in complex with STIL-CC | http://www.rcsb.org/pdb/explore/explore.do?structureId=4YYP | Publicly available at the RCSB Protein Data Bank (accession no. 4YYP). |
| Böhm R, Arquint C, Gabryjonczyk AM, Imseng S, Sauer E, Nigg E, Maier T, Hiller S | 2015 | STIL binding to Polo-box 3 of PLK4 regulates centriole duplication - NMR solution structure of human Polo-box 3 | http://www.bmrb.wisc.edu/data_library/summary/index.php?bmrbId=25552 | Publicly available at Biological Magnetic Resonance Data Bank (accession no. 25552). |
| Böhm R, Arquint C, Gabryjonczyk AM, Imseng S, Sauer E, Nigg E, Maier T, Hiller S | 2015 | STIL binding to Polo-box 3 of PLK4 regulates centriole duplication - Backbone assignment of human Polo-box 3 bound to STIL | http://www.bmrb.wisc.edu/data_library/summary/index.php?bmrbId=26547 | Publicly available at Biological Magnetic Resonance Data Bank (accession no. 26547). |

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
