## [Decision Letter]

Thank you for submitting your work entitled “STIL binding to Polo-box 3 of PLK4 regulates centriole duplication” for peer review at *eLife*. Your submission has been favorably evaluated by Tony Hunter (Senior Editor), a Reviewing Editor, and three reviewers.

The reviewers have discussed the reviews with one another and the Reviewing Editor has drafted this decision to help you prepare a revised submission.

Summary:

In this study Arquint et al. focus on the interaction between STIL and PLK4, two essential proteins involved at the earliest stages of centriole biogenesis. The authors elaborate on the interaction between STIL and PLK4 recently identified in several publications (Ohta et al., Nat. Comm, 2014; Kratz et al. Open Biol., 2015; Dzhindzhev et al., Curr. Biol., 2014). The authors use a biochemical approach to properly delineate the interaction domains of each protein. These experiments reveal that the coiled-coil (CC) domain of STIL is necessary and sufficient to interact with PLK4, and that the CC interacts with both the PB3 and L1 regions of Plk4. Most importantly, STIL-CC:PLK4-PB3 interaction was then explored in detail via biophysical methods. NMR showed that apo-PB3 was monomeric in solution whether bound to the STIL-CC or not. The PB3 polo box domain was folded from a continuous polypeptide chain, contrary to a previous study on murine PLK4 (Leung 2002), which reported that the PB3 is a domain-swapped dimer, despite 97% sequence identity. This contrasting result is very curious as both the earlier data and the data presented here seem solid.

The CC:PB3 interaction has nanomolar affinity with a 1:1 stoichiometry, and the authors obtain the structure of the STIL-CC:PLK4-PB3 complex by crystallography, showing the interaction mimics a conventional coiled-coil interaction. This interaction mechanism appears to be novel with respect to polo-boxes and their ligands. This structure enabled the authors to design mutations in STIL to dissect the function of the STIL:PB3 interaction, revealing that it is essential to promote centriole overduplication.

Essential revisions:

The reviewers would like you to re-write your paper to tone down some of your conclusions and to discuss the discrepancies with other published studies more fully. The following points should be addressed:

Please comment on the apparent discrepancy between Figure 1 (and related) depletion of STIL that results in more PLK4 at centrioles compared to Figure 3 where STIL-cc that is not at centrioles at all does not show more, but less PLK4 at centrioles.

Please discuss that [54] saw centrosomal localization of a PLK PB3 construct.

The data on the interaction between STIL cc and PLK4 L1 and its functional implications are rather weak. Although it is tempting to speculate on this, there are no strong data provided that would support claims that this interaction really occurs/is functionally relevant in-vivo. In addition it is not clear whether LI and CC could interact simultaneously with PB3.

Please comment on whether mutations can compromise specifically the PB3 interaction without affecting the self-interaction.

Please comment on the apparent discrepancy with Vulprecht at al., 2012, who could not see centriolar localization of a construct lacking the STAN motif (but containing the CC).

Please comment on the apparent discrepancy between Figure 4 and [65] who claim that PLK4 PB1 and 2, but not PB3, are required for STIL binding (Figure 1—figure supplement 1); [50] also show that PLK4 Polo Boxes are not sufficient for STIL binding.

Please make clear that you have not tested the requirement for centriole duplication, only centriole re-duplication.

In the Discussion please elaborate on the data in Figure 6. This figure is critical for the paper and the model that is raised in 7. However, most of the mutants that abrogate STIL-CC/ PLK4-PB3 interaction do not promote centriole amplification when overexpressed (Figure 6). However they still bind to (catalytic-domain+L1; 6D) and localize to the centriole (6F) suggesting that L1 is sufficient for their recruitment. A model (as in Figure 7) where STIL first binds to PB3 may not make sense, since that interaction is not needed for STIL localization? Does STIL localize to the centriole if the whole PB3 is removed? It is not clear why these STIL “M” mutants are not “active”; this is critical for the interpretation of the results and the model in Figure 7. (For example: is CC-PB3 interaction needed for SAS6 recruitment? Is it needed for PLK4 activation? Is PLK4 able to phosphorylate this mutant? Did the authors look in more detail at STIL-CC-PLK4 L1 binding as they did for PLK4 PB3 in Figure 4? Did they measure the binding kinetics of PLK4 L1 and STIL CC? If so, how does this compare to PB3? Are the interactions allosteric or competitive?)

Please discuss the following point. The authors propose that PLK4/STIL interaction mediates PLK4 activation. What is shown is that the interaction is required for centriole biogenesis, but activity is never directly measured. The authors propose that since the kinase domain is not involved in the interaction, binding and phosphorylation may therefore be independent. However, [65] showed that only the active kinase is able to bind STIL efficiently, suggesting therefore that phosphorylation and binding may be coupled. It would be interesting to discuss whether the interaction is dependent on phosphorylation or not; perhaps this is something the authors can infer from the structure.

---

## [Author Response]

*Please comment on the apparent discrepancy between*
Figure 1
*(and related) depletion of STIL that results in more PLK4 at centrioles compared to*
Figure 3
*where STIL-cc that is not at centrioles at all does not show more, but less PLK4 at centrioles*.

These two experiments differ in regard to presence or absence of endogenous STIL. Figure 1 depicts depletion of endogenous STIL, which results in significant accumulation of PLK4 both around centrioles and in the cytoplasm (increase of PLK4 levels around centrioles upon depletion of STIL has also been observed by [65], a reference to this publication has now been added). In the experiment shown in Figure 3 however, endogenous STIL was not depleted, hence no increase/decrease in PLK4 levels at centrioles is to be expected. Indeed, endogenous PLK4 levels are comparable to untreated cells and we therefore conclude that STIL CC overexpression has no effect on endogenous PLK4 localisation.

*Please discuss that*
[54]
*saw centrosomal localization of a PLK PB3 construct*.

Modified the manuscript accordingly (Introduction, fourth paragraph).

*The data on the interaction between STIL cc and PLK4 L1 and its functional implications are rather weak. Although it is tempting to speculate on this, there are no strong data provided that would support claims that this interaction really occurs/is functionally relevant in-vivo. In addition it is not clear whether LI and CC could interact simultaneously with PB3*.

[47] show that PLK4 does autoinhibit its own kinase activity, and interestingly, both the L1 linker and the PB3 domain are involved in this process. The L1 linker is required for the inhibition of kinase activity itself, whereas the PB3 domain seems to be required for relief of autoinhibition. Most interestingly however, direct interaction of PB3 with the L1 domain has not been observed, suggesting that a yet unidentified binding partner might be required to mediate such an interaction. While the above study has been performed with *Drosophila* PLK4, most recently, [60] also observed for human PLK4 that direct binding of STIL activates PLK4 by promoting self-phosphorylation of the kinase activation loop. In combination with our in vivo data indicating that PLK4 accumulates in an inactive form when STIL is depleted from cells, we are confident that the STIL CC-L1 interaction is highly relevant for PLK4 regulation, and therefore centriole duplication. Furthermore, we and others (65) show that STIL stabilizes PLK4 at centrioles. As active PLK4 destroys itself quickly via its phosphodegron located on the L1 linker region, stabilization might be achieved via a masking of the DSG motif upon STIL binding to L1 and a concomitant block in DSG recognition by βTrCP. Direct experimental confirmation of these points is beyond the scope of this study, but is subject of ongoing work in our labs.

We currently do not know whether STIL CC can simultaneously interact with L1 and PB3, but given their distinct binding interfaces, such a scenario is plausible. Especially, structural characterization of the binding of the L1 linker region to the STIL CC will be required to make further conclusions. This will be difficult, since our attempts to further narrow down the L1 binding region were unsuccessful so far, suggesting a non-linear folded binding region.

*Please comment on whether mutations can compromise specifically the PB3 interaction without affecting the self-interaction*.

As the mutated residues are the same ones as predicted to be involved in coiled-coil formation in the STIL oligomer, it is likely that STIL-CC/PLK4-PB3 binding and the STIL self-interaction are coupled and not independent events. The manuscript has been modified accordingly (Discussion, fourth paragraph).

*Please comment on the apparent discrepancy with Vulprecht at al., 2012, who could not see centriolar localization of a construct lacking the STAN motif (but containing the CC)*.

Deletion of the CC domain strongly disturbs centriolar localization, suggesting that it plays a dominant role in STIL localization via PLK4-mediated recruitment. In line with this, a clean deletion of the STAN domain does not significantly perturb centriolar localization of STIL (as also shown by [65], see Figure 8). However, truncation of STIL at amino acid 1060 (which removes the STAN domain plus some downstream C-terminal residues), interferes with correct localization (as shown by [86] and confirmed in Arquint et al., 2014), whereas removal of only the C-terminal downstream residues does not perturb centriolar localization nor interfere with the ability of STIL to cause robust centriole amplification (Arquint et al., 2014).

Therefore, presence of either the STAN domain or the downstream C-terminal residues seems to be necessary for stable localization of STIL to centrioles after initial recruitment. In addition, a recent study has shown that the Ana2 CC domain itself does not localize to centrioles in *Drosophila* embryos (18). We therefore conclude that interaction with PLK4 alone is not sufficient to stably integrate STIL into the cartwheel structure after it has been recruited to centrioles in a PLK4-dependent manner. We have clarified these points in the manuscript accordingly (Discussion,sixth paragraph).

Author response image 1.**DOI:**
http://dx.doi.org/10.7554/eLife.07888.023

*Please comment on the apparent discrepancy between*
Figure 4
*and*
[65]
*who claim that PLK4 PB1 and 2, but not PB3, are required for STIL binding (*Figure 1—figure supplement 1*);*
[50]
*also show that PLK4 Polo Boxes are not sufficient for STIL binding*.

[65] indeed see a reduction in STIL binding when using delta PB1 and delta PB2 PLK4 mutants in their Co-IP experiments. PB1 and 2 of PLK4 have been shown to be required for PLK4 dimerization (47), therefore, loss of structural integrity with those PLK4 mutants might explain a reduction in STIL binding. Furthermore, it is well established that PB1 and PB2 domains of PLK4 cooperate with Cep192/Cep152 in PLK4 recruitment to centrioles (66, 75). Therefore, the inability of those mutants to localize to the centriole might further influence the ability of PLK4 to bind to STIL. Also, it is not surprising to find an interaction between STIL and PLK4 ΔPB3, as the L1 linker region might be sufficient to maintain the binding.

On the other hand, Kratz et al. see no binding to STIL with PLK4 truncations that contain the PB3, which is a clear discrepancy to our data. We do not know why the interaction has been missed in that case. Given our in depth structural and biophysical characterization of the binding between PB3 and STIL CC we are very confident that the interaction described here is existent both in vivo and vitro.

*Please make clear that you have not tested the requirement for centriole duplication, only centriole re-duplication*.

We agree and have changed our Results section accordingly. (“ Having established the importance of the CC motif for the PLK4/STIL interaction, we next tested the requirement of this motif for STIL functionality in centriole reduplication”).

*In the Discussion please elaborate on the data in*
Figure 6*. This figure is critical for the paper and the model that is raised in 7. However, most of the mutants that abrogate STIL-CC/ PLK4-PB3 interaction do not promote centriole amplification when overexpressed (*Figure 6*). However they still bind to (catalytic-domain+L1; 6D) and localize to the centriole (6F) suggesting that L1 is sufficient for their recruitment*. *A model (as in*
Figure 7*) where STIL first binds to PB3 may not make sense, since that interaction is not needed for STIL localization?*

Centriole duplication requires STIL recruitment to centrioles and phosphorylation of the STIL STAN domain by PLK4, which then allows for SAS-6 binding, resulting in formation of a cartwheel structure. Both L1 and PB3 bind STIL. It is thus tempting to speculate that STIL is the factor regulating the relief of L1-mediated autoinhibition (47). We cannot determine a strict order of events between PB3 and L1 binding (we have modified Figure 7 to make this more clear). It is beyond the scope of this work to look at this in more detail, but it will be interesting to address this point in the future.

*Does STIL localize to the centriole if the whole PB3 is removed? It is not clear why these STIL* “*M*” *mutants are not* “*active*”*; this is critical for the interpretation of the results and the model in*
Figure 7.

We agree, there might be several reasons why the STIL M mutants are not active: reduced recruitment to PLK4/no PLK4 activation/loss of STIL oligomerisation.

See also reply to the third comment.

*(For example: is CC-PB3 interaction needed for SAS6 recruitment? Is it needed for PLK4 activation? Is PLK4 able to phosphorylate this mutant? Did the authors look in more detail at STIL-CC-PLK4 L1 binding as they did for PLK4 PB3 in*
Figure 4*? Did they measure the binding kinetics of PLK4 L1 and STIL CC? If so, how does this compare to PB3? Are the interactions allosteric or competitive*?*)*

We were unable to narrow down the L1 binding region, suggesting a non-linear folded segment. This hindered us to obtain a stably expressing protein fragment for further binding experiments and biophysical studies.

*Please discuss the following point. The authors propose that PLK4/STIL interaction mediates PLK4 activation. What is shown is that the interaction is required for centriole biogenesis, but activity is never directly measured*.

For our proposed mechanism, we build on our observations, that PLK4 levels accumulate in the absence of STIL (to the same degree as if PLK4 is protected against degraded) and further refer to [47]. Most importantly, in line with our proposed mechanism, it has very recently been shown by [60] that STIL activates PLK4 kinase activity in human cells. In vitro assays in this system are complicated due to problems in producing stable full-length PLK4.

*The authors propose that since the kinase domain is not involved in the interaction, binding and phosphorylation may therefore be independent*.

We resolve interactions of STIL and the PLK4-PB3 domain and further observe binding of STIL-CC to the L1 linker. In line with recent publications on PLK4 activation (47; 60), we hypothesize that STIL-CC binding to L1 could be implicated in relieving kinase autoinhibition. While the kinase domain is not sufficient for STIL-CC binding, it might still contribute to or influence STIL-CC PLK4-L1 interactions, such effects have not been excluded in our manuscript.

*However,*
[65]
*showed that only the active kinase is able to bind STIL efficiently, suggesting therefore that phosphorylation and binding may be coupled. It would be interesting to discuss whether the interaction is dependent on phosphorylation or not; perhaps this is something the authors can infer from the structure*.

Unfortunately, the current structural data don’t provide any information on this aspect, as the binding mode of STIL-CC to secondary sites in PLK4 remains unknown.